# Combined Endovenous Laser and Mechanochemical Ablation to Reduce Sclerosant for Three or Four Veins with Chronic Venous Insufficiency

**Kangjoo Choi [1]** , **Yujin Kwon [2]** , **Heejae Jun [3]** and **Myunghee Yoon [4],***

1 Department of Cardiovascular Surgery, Khatlon State Medical University, Dangara 735320, Tajikistan
2 Department of Surgery, Seoul Medical Center, Seoul 02053, Republic of Korea
3 Department of Vascular Surgery, Jun's Vascular Clinic, Busan 47256, Republic of Korea
4 Department of Surgery, Pusan National University, College of Medicine, Pusan National University Hospital, Busan 49241, Republic of Korea
* Correspondence: ymh@pusan.ac.kr; Tel.: +82-51-240-7238; Fax: +82-51-247-1365

**Abstract:** Background: A large amount of sclerosant is needed for the treatment of saphenous vein insufficiency with mechanochemical ablation (MOCA) for three or four veins with chronic venous insufficiency. In addition, what constitutes a safe amount is not clearly defined. In this study, we evaluate the feasibility of the combined endovenous laser and mechanochemical ablation to reduce the amount of sclerosant as compared with mechanochemical ablation monotherapy. Methods: A total of 327 patients diagnosed with superficial vein insufficiency between June 2018 and May 2020 and treated in a single center by one surgeon were evaluated retrospectively. There were 130 patients included who were treated with mechanochemical ablation (MOCA, Group I) and 197 patients who were treated with combined endovenous laser ablation and mechanochemical ablation (EVLA and MOCA, Group II). Results: The amount of sodium tetradecyl sulfate (STD) used per number of limbs was $5.5 \pm 2.05$ mL in Group I and $4.51 \pm 1.2$ mL in Group II ($p < 0.001$). The amount of STD used per number of veins was $4.77 \pm 1.91$ mL versus $3.12 \pm 1.02$ mL in Groups I and II, respectively, ($p < 0.001$). Recanalization rates within 52 weeks were 0% (0/130) in Group I and 5.58% (11/197) in Group II, while after 52 weeks they were 2.31% (3/130) in Group I and 6.60% (13/197) Group II and were not statistically significant. Complications within 4 weeks were 3.84% and 7.11% in Groups I and II, respectively. Conclusions: The results of this study show that combined EVLA and MOCA reduces the amount of sclerosant per the number of veins and legs treated as compared with MOCA monotherapy for three or four veins with chronic venous insufficiency. The combined EVLA and MOCA treatment in patients with three or four varicose veins has few side effects, reduces the amount of sclerotic agent, and can be considered to be an effective treatment method for inducing venous occlusion.

**Keywords:** chronic venous insufficiency; thermal endovenous laser ablation; non-thermal mechanochemical ablation; sclerosant; recanalization

## 1. Introduction

Chronic superficial venous insufficiency is a common disease worldwide which results in decreased quality of life (QOL) in patients who are affected. In the past, the aim of treatment was surgical removal of the pathological vein. However, recently, due to the advantages of less pain and wound complications, better cosmetic result, and early return to work, endovenous venous treatment has gained popularity in treating limbs with venous insufficiency. Endovenous treatments include endovenous thermal ablation (EVTA) with a laser (endovenous laser ablation, EVLA) or radiofrequent (radiofrequent ablation, RFA) device and ultrasound guided sclerotherapy using sclerosant injection or mechanochemical ablation (MOCA).

Previous studies that have compared EVLA, surgery, and foam sclerotherapy, have observed inferiority in reccurence rate and 5-year quality of life in a foam sclerotherapy group, despite its advantages with respect to less pain [1]. Another study that compared EVLT, RFA, surgery, and foam sclerotherapy, found that RFA and foam slcerothepapy had better postoperative quality of life with less postoperative pain [2]. EVTA has its weakness in using tumescence during the operation and a possible risk of heat-related skin or nerve injury. MOCA with the ClariVien® Occlusion Catheter system (Bridgemedica LLC, Boston, MA, USA) has been developed to solve these problems.

MOCA avoids thermal injury, but a considerable amount of sclerosant is needed to close the proximal portion of the greater saphenous vein (GSV). It is not clearly known how much sclerosant is safe in order not to induce side effects such as thrombotic complications. Thrombotic occlusion is fibrosis and ablation of blood vessels. Adverse effects associated with sodium tetradecyl sulfate (STD) therapy for treatment of leg veins include pain, urticaria, ulceration, hyperpigmentation, cutaneous necrosis, and telangiectatic matting. Other complications are pulmonary embolism, anaphylaxis, and venous thrombosis [3–5]. Therefore, the reduced usage of sclerosant is desirable to prevent unfavorable events. A modified procedure can be applied on the basis of this viewpoint.

The aim of this study was to evaluate the safety and feasibility of concomitant thermal and nonthermal endovenous ablation for patients who mostly had three or more superficial vein insufficiency and to analyze the amount of sclerosant as compared with MOCA monotherapy.

## 2. Methods

Between June 2018 and May 2020, a total of 327 patients diagnosed with superficial vein insufficiency and treated by one surgeon at a single center were evaluated retrospectively. According to the treatment methods, the patients were grouped into two groups: Group I included 130 patients who were treated the mechanochemical ablation (MOCA) monotherapy, Group II included 197 patients who were treated with the combined endovenous laser and mechanochemical ablation (EVLA with MOCA) treatment. The patients who were selected for the combined treatment (Group II) mostly had insufficiencies in more than three veins and were intended to be candidates in order to reduce the amount of the sclerosant.

Included were patients aged between 18 and 75 years with symptomatic varicose veins, Clinical Etiologic Anatomic Pathophysiologic (CEAP) class C2–6, and GSV incompetence defined by a reflux time of more than 0.5 s on duplex imaging [6]. The patients were examined in the standing position, and reflux was measured after manual compression and release of the calf and thigh. As bilateral treatment was permitted, both legs had the same treatment during the same operation. Patients with recurrent varicose veins were also included if the GSV was preserved to the groin on duplex imaging.

A sedative and analgesic (midazolam) were administered intravenously before the procedure in most patients.

In Group II (EVLA + MOCA), the EVLA procedure was started near the saphenofemoral junction (SFJ) and was terminated at the lower thigh in the GSV and the MOCA procedure was performed below the knee and in the small saphenous vein (SSV).

The EVLA procedure was performed under duplex guidance with a 1940 nm GaAIAs laser diode, continuous wave, 600 micrometer bare-tip fiber (Atoven-1, Diotec, Korea) for all patients. The laser fiber was advanced until 2 cm below the saphenofemoral junction (SFJ) and the GSV was ablated during withdrawal of the fiber at 10 mm per 7–8 s with 4–7 watt of laser energy. All treatments were performed in an operation room under tumescent local anesthesia, using a normal saline solution of 0.1 percent lidocaine (N/S 500 mL + 2% Lidocaine 25 mL). The solution was administered using an infusion pump (Endo-jet, Mesa Medical, Korea) under ultrasound guidance and the aim was to administer 10 mL per cm GSV tumescent anesthesia in Group II.

MOCA with Clarivein® (Bridgemedica LLC, Boston, MA, USA) was performed with two steps of action; mechanical destruction of the vessel endothelium by a rotating catheter tip and a sclerosant spray from the tip of the catheter to ensure maximal effect. No tumescent anaesthesia was required for this procedure. All the patients were in a supine position with the leg slightly flexed and abducted to enhance the access to both the GSV and small saphenous vein (SSV). A Seldinger technique was used to introduce a 5 Fr introducer sheath into either GSV or SSV under ultrasound guidance and flushed with saline. The ClariVein® OC infusion catheter (Bridgemedica LLC, Boston, MA, USA) tip was inserted through the sheath and the tip of the wire was positioned 10 mm distal to the saphenofemoral junction or saphenopopliteal junction. The sheath was withdrawn to just beyond the puncture site to prevent activation of the probe within the sheath. Wire rotation was activated for a few seconds to induce spasm of the proximal vein, and then while the wire continued to rotate, infusion of the sclerosant was started simultaneously with catheter pullback. The activated catheter was steadily withdrawn at 1 cm every 7 to 10 s. The sclerosant used was 2.0% liquid sodium tetradecyl sulphate (STD) [7].

Generally, 0.1–0.2 mL of sclerosant was injected every 1 cm pullback on the catheter. A completion duplex ultrasound was performed after the procedure to confirm the patency of the common femoral vein and the deep venous system. Vein diameter was measured at the widest part of the vein treated excluding the first 2 cm of the vein.

After the procedure, the leg was wrapped in sterile absorbent bandages and covered with a cohesive compression bandage for 48 h. Then, patients were instructed to use a compression stocking to the groin for 2 weeks. No specific analgesia was prescribed. All patients were encouraged to resume work and normal activity as soon as they were able.

The treatment cost for MOCA alone is the same as for combined MOCA and EVLA surgery. EVLA surgery fees are not additionally calculated according to the characteristics of the national insurance system.

The study design was approved by the Institutional Review Board of the Pusan National University Hospital (PNUH No. 2109-011-107) and was conducted in accordance with the Declaration of Helsinki.

### 2.1. Follow-Up

Patients were asked to document the level of peri- and post-procedural pain. A median follow-up period was 1 year. Patients visited the clinic after two weeks, two months, six months, and one year, and an ultrasound study and clinical exam were performed. Color duplex scan was performed, scanning the full length of the treated vein, and compressibility and reflux of the vein were tested. A successfully obliterated vein was solid with no visible lumen and could not be compressed, without flow on color duplex and Valsalva.

### 2.2. Statistical Analysis

Continuous variables are reported as mean and standard deviation and categorical variables as absolute number and percent, unless stated otherwise. Continuous data were compared using the Student *t*-test for parametric and non-parametric data. Categorical data were compared using the chi-square or Fisher exact tests. Statistical significance was assumed at $p < 0.05$, and the statistical analyses were performed using the language R (http://cran.r-project.org, accessed on 22 December 2022) version 3.6.0. and additional packages (stats, forest plot, ROC).

### 3. Results

Clinical characteristics are described in Table 1. The mean ages were $51.89 \pm 12.95$ (Group I) and $54.96 \pm 12.53$ (Group II) and there was 74.6% and 62.9% females in Groups I and II, respectively. According to CEAP Classification, C2–C4 were 99.23% (129/130) in Group I and 98.47% (194/197) in Group II. The diameter of the GSV was $72 \pm 2.27$ mm in Group 1 and $8.63 \pm 2.77$ mm in Group II, which was statistically significant ($p < 0.001$).

Patients with varicose veins in both legs were 46.15% (60/130) and 90.86% (179/197) in Groups I and II, respectively (*p* < 0.001).

**Table 1.** Clinical characteristics between MOCA (Group I) and combined EVLA and MOCA (Group II).

| Variables | Overall (*n* = 327) | Group I (*n* = 130) | Group II (*n* = 197) | *p*-Value |
|---|---|---|---|---|
| Age | 53.74 ± 12.77 | 51.89 ± 12.95 | 54.96 ± 12.53 | 0.034 |
| Female/Male (%) | 221 (67.6)/106 (32.4) | 97 (74.6)/33 (25.4) | 124 (62.9)/73 (37.1) | 0.037 |
| CEAP Classification | | | | |
| C2 | 163 (49.85) | 63 (48.46) | 100 (50.76) | |
| C3 | 140 (42.81) | 61 (46.92) | 79 (40.10) | |
| C4 | 20 (6.12) | 5 (3.85) | 15 (7.61) | 0.385 |
| C5 | 2 (0.61) | 0 (0) | 2 (1.02) | |
| C6 | 2 (0.61) | 1 (0.77) | 1 (0.51) | |
| Diameter GSV (mm) | 8.27 ± 2.62 | 7.72 ± 2.27 | 8.63 ± 2.77 | <0.001 |

MOCA, mechanochemical ablation; EVLA, endovenous laser ablation; CEAP, Clinical Etiologic Anatomic Pathophysiologic; C2, CEAP classification; GSV, greater saphenous vein.

Patients with three or more varicose veins were 15.40% (20/130) in Group I and 69.03% (136/197) in Group II (*p* < 0.001) (Table 2).

**Table 2.** Number of veins between MOCA (Group I) and combined EVLA and MOCA (Group II).

| Number of Veins (%) | Overall (*n* = 327) | Group I (*n* = 130) | Group II (*n* = 197) | *p*-Value |
|---|---|---|---|---|
| 1 | 68 (20.79) | 55 (42.3) | 13 (6.6) | |
| 2 | 103 (31.50) | 55 (42.3) | 48 (24.37) | <0.001 |
| 3 | 98 (29.97) | 14 (10.78) | 84 (42.64) | |
| 4 | 58 (17.74) | 6 (4.62) | 52 (26.39) | |

Total amount of STD used was 7.56 ± 2.71 mL in Group I and 8.52 ± 2.54 mL in Group II (*p* < 0.001), and the amount of STD used per number of legs was 5.5 ± 2.05 mL in Group I and 4.51 ± 1.2 mL in Group II (*p* < 0.001). The amount of STD used per number of veins was 4.77 ± 1.91 mL and 3.12 ± 1.02 mL in Groups I and II, respectively (*p* < 0.001) (Table 3).

**Table 3.** Sclerosant amounts between MOCA (Group I) and combined EVLA and MOCA (Group II).

| Variables | Overall (*n* = 327) | Group I (*n* = 130) | Group II (*n* = 197) | *p*-Value |
|---|---|---|---|---|
| Total STD (mL) | 8.14 ± 2.64 | 7.56 ± 2.71 | 8.52 ± 2.54 | <0.001 |
| STD/Leg (mL) | 4.91 ± 1.69 | 5.5 ± 2.05 | 4.51 ± 1.27 | <0.001 |
| STD/Vein (mL) | 3.77 ± 1.65 | 4.77 ± 1.91 | 3.12 ± 1.02 | <0.001 |

STD: sodium tetradecyl sulfate.

Complication rates were 3.84% vs. 7.11% in Groups I and II, respectively (Table 4). Recanalization was 0% (0/130) within 52 weeks and 2.31% (3/130) after 52 weeks in Group I, while it was 5.58% (11/197) within 52 weeks and 6.60% (13/197) after 52 weeks in Group II and it was not statistically significant (*p* = 0.247), whereas, the total rate of recanalization was 2.31% (3/130) in Group I and 12.18% (24/197) in Group II, which was statistically significant (*p* < 0.003) (Table 5). For recanalization after combined surgery, there were three cases of recurrence in the proximal portion of both GSVs (Table 6).

**Table 4.** Type of complications between MOCA (Group I) and combined EVLA and MOCA (Group II).

| Type of Complication | Group I (*n* = 130) | Group II (*n* = 197) *p*-Value |
| --- | --- | --- |
| | Within 4 Weeks | Within 4 Weeks |
| Edema | 0 | 2 |
| Hemoglobinuria | 1 | 0 |
| Local skin infection | 0 | 1 |
| Pain along vein | 0 | 1 |
| Pigmentation | 2 | 1 |
| Rash, thrombophlebitis | 2 | 8 |
| Wound oozing | 0 | 1 |
| Total (%) | 5 (3.85) | 14 (7.11) 0.765 |

**Table 5.** Recanalization rates between MOCA (Group I) and combined EVLA and MOCA (Group II).

| Recanalization (%) | Overall (*n* = 327) | Group I (*n* = 130) | Group II (*n* = 197) | *p*-Value |
| --- | --- | --- | --- | --- |
| Early (within 52 weeks) | 11 (3.36) | 0 (0) | 11 (5.58) | 0.247 |
| Delayed (after 52 weeks) | 16 (4.89) | 3 (2.31) | 13 (6.60) | |
| Total (%) | 27 (8.25) | 3 (2.31) | 24 (12.18) | 0.003 |

**Table 6.** Type of recanalization between MOCA (Group I) and combined EVLA and MOCA (Group II).

| Type of Recanalization | Group I (*n* = 130) | | Group II (*n* = 197) | |
| --- | --- | --- | --- | --- |
| | Early | Delayed | Early | Delayed |
| Both GSV prox | 0 | 0 | 3 | 1 |
| Both GSV thigh | 0 | 0 | 1 | 0 |
| Both SSV prox | 0 | 1 | 0 | 0 |
| Lt GSV bk | 0 | 0 | 0 | 1 |
| Lt GSV thigh | 0 | 0 | 0 | 3 |
| Lt GSV thigh, Lt SSV | 0 | 0 | 0 | 1 |
| Lt SFJ | 0 | 1 | 0 | 0 |
| Lt SSV | 0 | 0 | 2 | 2 |
| Lt SSV, GSV bk | 0 | 0 | 1 | 0 |
| Rt GSV bk, Lt GSV thigh | 0 | 0 | 0 | 1 |
| Rt GSV thigh | 0 | 0 | 3 | 1 |
| Rt GSV, Lt SSV | 0 | 1 | 0 | 0 |
| Rt SSV | 0 | 0 | 1 | 3 |
| Total (%) | 0 (0) | 3 (2.31) | 11 (5.58) | 13 (6.60) |

MOCA, mechanochemical ablation; EVLA, endovenous laser ablation; GSV, great saphenous vein; SSV, small saphenous vein; Lt, left; SFJ, saphenofemoral junction; bk, below knee.

## 4. Discussion

This study was retrospectively performed; therefore, the classified subjects had many limitations, but there was little bias because one vascular surgeon performed the operation.

In this study, MOCA monotherapy and combined EVLA and MOCA treatment were compared. The combined EVLA and MOCA treatment was mostly performed for patients who had developed varicose veins in three or more veins and the occlusion rate of insufficient varicose vein was high at 1-year follow up after the procedure. The amount of STD used was also reduced per number of limbs and per number of veins. In addition, bilateral procedures could be successfully performed for multiple veins. There were no incidences of sclerosant-induced deep vein thrombosis (DVT) or endothermal heat-induced DVT observed in both groups.

MOCA using Clarivein® is widely performed for its benefits on ablating long and short saphenous vein with insufficiency on a walk-in-walk-out basis without the need for

tumescent anesthesia. However, the maximal sclerosant volume depends on a patient's weight when using polidocanol. There is no evidence of treating a larger GSV (>12 mm) with MOCA.

The volume of sclerosant used was determined case by case, examination with duplex ultrasound evaluating mechanical and chemical destruction of the vein resulting in collapse of the vein. It is important to tract the amount of the total sclerosant used so that it does not exceed the safety dose. Generally, 0.1–0.2 mL of sclerosant is injected for every 1 cm pullback on the catheter. The maximum recommended dose of STD used for a procedure should not exceed 10 mL of 3%, which is 30 mg/mL and if using 2% STD, this is 15 mL [8] and for this reason, the extent of vein treated can be limited when using higher concentrations.

For varicose veins with three or four veins with chronic venous insufficiency, MOCA monotherapy alone could not be used because of the high burden of the sclerosing agent dose.

Therefore, GSV was treated with EVLA and MOCA was performed just below the knee to reduce the sclerosing agent and effectively occlude the varicose veins.

Even in patients with four varicose veins, the total volume of sclerosing agent did not exceed 10 mL, and by combining EVLA with MOCA, complications such as nerve damage and skin side effects were few.

Lam et al. reported that post-treatment duplex at 6 weeks follow-up exam showed 100%, 96.4%, and 56.5% vein occlusion rate using 2% and 3% polidocanol liquid and 1% polidocanol microfoams during the MOCA procedure, respectively [9]. After MOCA using ClariVein, 3.3% vein obliteration rate was observed eight weeks after the procedure. In a recent randomized controlled trial, a comparison of MOCA with clarivein was performed. The procedures both showed early return to work and low pain, but axial recanalization was higher in MOCA at 1-year follow-up [1,10,11].

In our study, Groups I and II both showed high efficacy in treating superficial vein insufficiency. The groups both showed significant decreases in severity of disease, and low procedural pain with short recovery times. Recanalization was 0% within 52 weeks and 2.31% after 52 weeks in Group I, while it was 5.58% within 52 weeks and 6.60% after 52 weeks in Group II and it was not statistically significant ($p = 0.247$), whereas, total rate of recanalization was 2.31% in Group I and 12.18% in Group II, which was statistically significant ($p < 0.003$). For recanalization after combined surgery, there were three cases of recurrence in the proximal portion of both GSVs. After confirming a case of recanalization of proximal GSV, the amount of watts of laser treatment was increased.

In Group II, 24 patients had recanalization; 16 patients were treated with 4 watts, 7 patients were treated with 5 watts, and 1 patient was treated with 6 watts laser energy. The higher total recanalization rate of Group II seemed to be due to the lower energy of laser in the early experiences. Since May 2019, 7 watts laser energy was used during the procedure, and there was no recanalization found. For EVTA, tumescent anesthesia is required and there is direct pain from the injection, and the patients are more prone to thermal injuries to the skin and the nerve [12,13]. The benefits of MOCA include unnecessary tumescent anesthesia and a lack of heat-related complications [7,14] in the GSVand also in the SSV [15,16]. In addition, MOCA has been shown to be superior with respect to less procedural time and intra/post-procedural pain than RFA and EVLT [17,18]. The limitations on the extent of limbs and veins treated owing to the safe dose of STD can be overcome by using combined EVLA and MOCA treatment.

There were no differences with respect to return to normal activities and work between the two groups.

There were few complications of skin reaction, venous pain, and phlebitis in either the MOCA alone or the combined EVLA and MOCA groups, and there was no significant difference statistically.

The combined EVLA and MOCA treatment in patients with three or four varicose veins has few side effects, reduces the amount of sclerotic agent, and is considered to be an effective treatment method for inducing venous occlusion.

## 5. Limitations

The two groups were not equal in age, sex, diameter of GSV, and number of veins treated. In Group II, the patients had more bilateral treatment than in Group I. It seems that Group II was more severe in pathology and needed more sclerosant than patients Group I. The study omitted the details of follow-up and simply compared the results at 1 year (52 week).

## 6. Conclusions

The results of this study have shown that combined EVLA and MOCA treatment reduces the amount of sclerosant per number of veins and legs treated as compared with MOCA monotherapy for three or four veins with chronic venous insufficiency. The combined EVLA and MOCA treatment in patients with three or four varicose veins has few side effects, reduces the amount of sclerotic agent, and is considered to be an effective treatment method for inducing venous occlusion.

**Author Contributions:** Conceptualization, K.C., H.J. and M.Y.; operation and methodology, H.J.; data collection and processing, K.C., M.Y., Y.K.; formal analysis, M.Y.; literature search, K.C., H.J., Y.K. and M.Y.; writing—original draft preparation, K.C. and M.Y.; writing—review and editing, M.Y. H.J. is a co-author: is a doctor who performed varicose vein surgery at Jun's Vascular Clinic. He will report their results after surgery. The first author: K.C., is a professor at the Medical University of Tajikistan. During the sabbatical period, he worked at Jun's Vascular clinic and contributed to the data collection and analysis of this paper. There is no patient's personal information of Jun's Vascular clinic in the ethics committee of Pusan National University Hospital. So, the data was approved by the Ethics Committee. All authors have read and agreed to the published version of the manuscript.

**Funding:** This research received no external funding.

**Institutional Review Board Statement:** This study was conducted according to the guidelines of the Declaration of Helsinki. The study design was approved by the Institutional Review Board of Pusan National University Hospital (PNUH No. 2109-011-107).

**Informed Consent Statement:** Informed consent was obtained from all subjects involved in the study.

**Data Availability Statement:** Regarding the sharing of research data, it can be shared with the corresponding author(ymh@pusan.ac.kr) in compliance with the principle of non-infringement of personal information.

**Acknowledgments:** This work was supported by a 2-year Research Grant of Pusan National University.

**Conflicts of Interest:** The authors declare no conflict of interest.

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
