# Peer review of "Combined Endovenous Laser and Mechanochemical Ablation to Reduce Sclerosant for Three or Four Veins with Chronic Venous Insufficiency"

_2813-2475, doi:10.3390/jvd2010004_

Round 1

Reviewer 1 Report

This is a retrospective analysis of the 327 patients diagnosed with superficial vein insufficiency between June 22 2018 and May 2020. The patient were treated in a single center by one surgeon.

The authors included 130 patients in nonthermal mechanochemical ablation (MOCA, Group I) and 197 patients in combined thermal and nonthermal endovenous laser ablation treatment (EVLA and 25 MOCA, Group II).

The authors show that the amount of 27 STD used per number of veins was 4.77 ± 1.91 mL versus 3.12 ± 1.02 mL in Group I and Group II, 28 respectively (p<0.001). Recanalization rates within 52 weeks were 0% (0/130) in Group I and 5.58% 29 (11/197) in Group II, while after 52 weeks they were 2.31% (3/130) in Group I and 6.60% (13/197) 30 Group II and were not statistically significant.

First of all, the recanalization difference between group I and II, seems significant despite the authors clam, 0 vs 5.58% or 2.31% vs 6.6%. They may need to reanalysis this.

While the title says “Combined thermal and mechanochemical ablation to reduce sclerosant for chronic venous insufficiency” and the results support this, the authors surprisingly have made any comment on this in the conclusion.

Author Response

First of all, the recanalization difference between group I and II, seems significant despite the authors clam, 0 vs 5.58% or 2.31% vs 6.6%. They may need to reanalysis this.

: Recanalization was 0% (0/130) within 52 weeks and 2.31% (3/130) after 52 weeks in Group I while 5.58% (11/197) within 52 weeks and 6.60% (13/197) after 52 weeks in Group II and it was not statistically significant (p=0.247). While total rate of recanalization was 2.31% (3/130) in Group I while 12.18% (24/197) in Group II, it was statistically significant (p<0.003) (Table 5).

While the title says “Combined thermal and mechanochemical ablation to reduce sclerosant for chronic venous insufficiency” and the results support this, the authors surprisingly have made any comment on this in the conclusion.

: The conclusion was modified.

: This study showed that the combined procedure (EVLA and MOCA) reduced the amount of sclerosant per the number of veins and legs treated, in comparison to the mono-therapy (MOCA). This study had some statistical limitations mentioned above. Nevertheless, it can be said that the combined endovenous ablation procedure had a trend to reduce the amount of sclerosant in patients with multi-vein insufficiency.

Reviewer 2 Report

how was it determined which patients get assigned to which group? this should be clarified in the methods section. also in group II, how was it decided which vein will be treated with EVLT and which vein will be treated with MOCA? 

it is unclear if in group II, the above knee GSV was treated with EVLT and below knee GSV was treated with MOCA? please clarify in manuscript. 

please include a limitations section - what are the limitations of your study? one major limitation is that both groups are not equal at baseline. Group 2 size of veins is larger, also group 2 has more bilateral treatments. 

what are the side effects of using too much sclerosant? please include a paragraph on that. 

a trend is noted towards what the authors state in the conclusions but it is not statistically significant. i would recommend revising the conclusions to mention that trends towards these things were observed but were not statistically significant etc. 

was any incidence of sclerosant induced DVT or endothermal heat induced DVT noted? either way, please mention that in the manuscript. 

Author Response

how was it determined which patients get assigned to which group? this should be clarified in the methods section.

: This study was designed retrospectively. The patients were divided into two groups according to the method of procedures. I added it briefly in the methods section as you recommended.

also in group II, how was it decided which vein will be treated with EVLT and which vein will be treated with MOCA?

: In group II, the selection of the procedure was not related to the vein.

it is unclear if in group II, the above knee GSV was treated with EVLT and below knee GSV was treated with MOCA? please clarify in manuscript.

: In group II, EVLT was applied to the proximal GSV near SFJ initially then applied down to lower thigh. MOCA was applied to GSV below knee and SSV. This is clarified in the manuscript.

please include a limitations section - what are the limitations of your study? one major limitation is that both groups are not equal at baseline. Group 2 size of veins is larger, also group 2 has more bilateral treatments.

: Yes, the limitations section is added and limitations are mentioned there.

what are the side effects of using too much sclerosant? please include a paragraph on that.

: The side effects of too much sclerosant are described in introduction.

a trend is noted towards what the authors state in the conclusions but it is not statistically significant. i would recommend revising the conclusions to mention that trends towards these things were observed but were not statistically significant etc.

: Some statistical limitations are noted and the conclusion is revised.

was any incidence of sclerosant induced DVT or endothermal heat induced DVT noted? either way, please mention that in the manuscript.

: What you mentioned is added in the manuscript. Thank you so much for your kind mentioning.

Reviewer 3 Report

I am unclear on precisely what question the authors are asking in the current work. 

I think that there are a number of issues with the English within the report, that require to be addressed.

In the Results section GSV vein diameter are recorded in cm rather than mm.

There is duplication of a number of references.

Author Response

I am unclear on precisely what question the authors are asking in the current work.

: Many sentences and contents are modified to improve communication more clearly.

I think that there are a number of issues with the English within the report, that require to be addressed.

: The manuscript was checked by native English-speaking expert.

In the Results section GSV vein diameter are recorded in cm rather than mm.

: cm unit in the results section is revised and added at Table 1.

There is duplication of a number of references.

: That is corrected.

Round 2

Reviewer 2 Report

the authors have revised their manuscript 

Author Response

The authors provide a rather large patient population (chronic venous insufficiency) and compare two different treatment strategies (MOCA versus EVLA and MOCA) with corresponding adequate statistical analyses. The tables are of high quality. Two clinical cases in the form of a case report - one for MOCA mono and one for EVLA and MOCA - would be nice so that your strategy can be seen visualized at a glance. Have you provided a cost analysis? Or at least a cost estimate for the materials you used? You don't need economic evaluation applying QUALYs, an overview of the material coasts would be fine ...

: The treatment cost for MOCA alone is the same as for combined MOCA and EVLA surgery.

EVLA surgery fees are not additionally calculated according to the characteristics of the national insurance system. .

Please indicate abbreviations before using them for the first time in the text.

 : The abbreviation was first indicated and used in the main text to correct it.

 : I changed title 

- Please use consistent terminology and use all definitions very clearly and strictly throughout the manuscript.

 : Revised text to use consistent terms and definitions

- Please explain all your statements and your perspective in more detail if necessary (e.g., Abstract: "It would be better to reduce the amount of sclerosant as much as possible." - why???)

 : Discussion

This study is a retrospectively performed, so the classified subjects have many limitations, but there is little bias because one vascular surgeon performed the operation.

In this study, MOCA mono-therapy and combined EVLA and MOCA therapy were compared. Combined EVLA and MOCA therapy was mostly performed for patient who developed varicose veins in three or more veins and the occlusion rate of insufficient varicose vein was high at 1-year follow up after the procedure. Also, the amount of STD used was reduced per number of limbs and per number of veins. In addition, bilateral procedures could be successfully performed for multiple veins. Neither any incidence of sclerosant induced DVT nor endothermal heat induced DVT were ob-served in both groups.

MOCA using Clarivein® is widely performed for its benefits on ablating long and short saphenous vein with insufficiency on a walk-in-walk-out basis without need of tumescent anesthesia. But, the maximal sclerosant volume depends on the patients’ weight when using polidocanol. There is no evidence of treating larger GSV (> 12 mm) with MOCA.

The volume of sclerosant used was determined case by case, examination with duplex ultrasound evaluating mechanical and chemical destruction of the vein resulting in collapse of the vein. It is important to tract the amount of the total sclerosant used so that it does not exceed the safety dose. Generally, 0.1 ml-0.2 ml of sclerosant is injected for every 1 cm pullback on the catheter. The maximum recommended dose of STD used for a procedure should not exceed 10 ml of 3%, which is 30 mg/ml and if using 2% STD, is 15 ml [8] and for this reason, extent of vein treated can be limited when using higher concentrations.

For varicose veins with 3 or 4 veins, MOCA mono-therapy alone could not be used because of the high burden of sclerosing agent dose.

Therefore, GSV was treated with EVLA and MOCA was performed just below the knee to reduce the sclerosing agent and effectively occlude varicose veins.

Even in patients with 4 varicose veins, the total volume of sclerosing agent did not exceed 10 ml, and by combining EVLA with MOCA, complications such as nerve dam-age and skin side effects were few.

Lam et al. reported that post-treatment duplex at 6 weeks follow-up exam showed 100%, 96.4% and 56.5% vein occlusion rate using 2% and 3% polidocanol liquid and 1% polidocanol microfoams during the MOCA procedure, respectably [9]. After MOCA using clarivein, 3.3% vein obliteration rate was observed eight weeks after the procedure. In recent randomized controlled trial, comparison of MOCA with clarivein was performed.  Both procedures showed early return to work and low pain, but axial re-canalization was higher n MOCA at 1-year follow up [1,10-11].

In our study, both Group I, and Group II showed high efficacy in treating superfi-cial vein insufficiency. Both groups showed significant decrease in severity of disease, low procedural pain with short recovery time. Recanalization was 0% within 52 weeks and 2.31% after 52 weeks in Group I while 5.58% within 52 weeks and 6.60% after 52 weeks in Group II and it was not statistically significant (p=0.247). While total rate of recanalization was 2.31% in Group I while 12.18% in Group II, it was statistically significant (p<0.003). For recanalization after combined surgery, there were three cases of recurrence in proximal portion of both GSV. After confirming a case of recanalization of proximal GSV, the amount of watt of laser treatment was increased.

In Group II, 24 patients had recanalization; 16 patients were treated with 4 watts, 7 with 5 watts and 1 patient with 6 watts laser energy. The higher total recanalization rate of group II seemed to be due to lower energy of laser in the early experiences. Since May 2019, 7 watts laser energy was used during the procedure, and there was no recanalization found. For EVTA, tumescent anesthesia is required and there is not only the direct pain from the injection, but also the patients are more prone to thermal injuries to the skin and the nerve [12,13]. MOCA shows benefits on unnecessity for the tumescent anesthesia and lack of heat related complications [14,15], not only in GSV, but also in SSV [16,17]. Also, MOCA was superior in less procedural time and in-tra/post-procedural pain than RFA and EVLT [18,19]. The limitations on the extent of limbs and veins treated owing to the safe dose of STD can be overcome by combined EVLA and MOCA therapy.

Return to normal activities and work was no difference between two groups.

There were few complications of skin reaction, venous pain, and phlebitis in either the MOCA alone or the combined EVLA and MOCA group, and there was no significant difference statistically.

Combined EVLA and MOCA treatment in patients with 3 or 4 varicose veins has few side effects, reduces the amount of sclerotic agent, and is considered to be an effective treatment method for inducing venous occlusion.
